# Adoption of Machine Learning in Pharmacometrics: An Overview of Recent Implementations and Their Considerations

**DOI:** 10.3390/pharmaceutics14091814

**Published:** 2022-08-29

**Authors:** Alexander Janssen, Frank C. Bennis, Ron A. A. Mathôt

**Affiliations:** 1Department of Clinical Pharmacology, Hospital Pharmacy, Amsterdam University Medical Center, 1105 Amsterdam, The Netherlands; 2Quantitative Data Analytics Group, Department of Computer Science, Vrije Universiteit Amsterdam, 1081 Amsterdam, The Netherlands

**Keywords:** machine learning, pharmacometrics, pharmacokinetics, pharmacodynamics

## Abstract

Pharmacometrics is a multidisciplinary field utilizing mathematical models of physiology, pharmacology, and disease to describe and quantify the interactions between medication and patient. As these models become more and more advanced, the need for advanced data analysis tools grows. Recently, there has been much interest in the adoption of machine learning (ML) algorithms. These algorithms offer strong function approximation capabilities and might reduce the time spent on model development. However, ML tools are not yet an integral part of the pharmacometrics workflow. The goal of this work is to discuss how ML algorithms have been applied in four stages of the pharmacometrics pipeline: data preparation, hypothesis generation, predictive modelling, and model validation. We will also discuss considerations before the use of ML algorithms with respect to each topic. We conclude by summarizing applications that hold potential for adoption by pharmacometricians.

## 1. Introduction

### 1.1. Background

Pharmacometrics is a multidisciplinary field utilizing mathematical models of physiology, pharmacology, and disease to describe and quantify the interactions between medication and patient. This involves models of drug pharmacokinetics (PK), pharmacodynamics (PD), exposure-response (PK/PD), and disease progression. One of the main themes of interest is the explanation of variability in drug response between patients. Various statistical techniques have been adopted to quantify such inter-individual variation (IIV) [1].

Non-linear mixed effect (NLME) modelling has been embraced as a statistical method for describing treatment effect on a population and individual level [2,3]. Population PK modelling makes efficient use of sparse data by pooling information of multiple individuals, and breaking down treatment response in shared and individual effects. Observations of the dependent variable (i.e., drug concentrations or treatment effect) can then be used to adapt the prediction to the individual patient, resulting in higher accuracy.

Recently, however, advances in hospital digitization, data collection, and inclusion of increasingly extensive laboratory testing in standard clinical care have resulted in the availability of richer datasets. This increased accessibility of complex data sources such as genomic or gene expression data stresses current modelling approaches as they can lack the flexibility to handle these data. As a response, more attention is being paid to the opportunity of using machine learning (ML) algorithms as an innovative strategy for pharmacometric modelling [4,5]. The field of ML has seen an explosive boost of promising applications for image analysis, text recognition, and other high-dimensional data. There are many examples of their successful application in the medical domain, for example for the diagnosis of breast cancer [6], identification of biomarkers from gene expression data [7], and survival analysis [8]. As ML methods offer strong predictive performance there is no denying that its adoption in pharmacometrics brings with it exciting new modelling opportunities.

As the relatively young ML research field is maturing at a rapid pace, more advanced model architectures are frequently being proposed in order to further improve predictive accuracy. Consequently, understanding the differences and intricacies of distinct learning methods is becoming increasingly more difficult for non-experts. A proper understanding of the advantages and pitfalls of these methods is essential for their responsible and reliable use, especially for clinical applications. As most of the emphasis has been put on the supposed high predictive accuracy of ML methods, it is easy to become overconfident in their abilities. It is thus important to monitor and guide the adoption of ML in pharmacometrics.

In this review, we will discuss recent approaches for the use of ML algorithms in the context of pharmacometrics, while also providing important considerations for their use. For some examples, we will provide demonstrations based on simulation experiments. We also discuss the important concept of model validation and the importance of understanding what is actually learned by the algorithm.

In this work, we will be assuming a general understanding of ML and the most common algorithms. For those wanting to learn more about the basic concepts of ML, Badillo et al. offer an excellent tutorial on ML aimed at pharmacometricians [9].

### 1.2. Structure of This Review

This review is structured as follows. First, we discuss applications of ML algorithms in three stages of the pharmacometrics pipeline: data preparation, hypothesis generation, and predictive modelling. We define these stages as follows: data preparation deals with the imputation of missing data and dimensionality reduction. Next, in the section on hypothesis generation we discuss methods for clustering data, and how ML can be used for the detection of influential covariates. In the predictive modelling section, we discuss ML-based alternatives to traditional modelling approaches. We will conclude our review of recent application with a discussion on model validation, focussing mainly on estimating model generalizability and the interpretation of ML models.

For each topic, we first discuss the current approach, its (possible) limitations, followed by what ML techniques have been proposed to address the issues. At the end of each topic, we will summarize the discussion with considerations for the use of ML for each issue.

### 1.3. Literature Search

In order to support the initial framing of our discussion we performed a literature search. Our objective was to find recent articles discussing ML in the context of pharmacometrics. The following search query for PubMed was constructed:

(“machine learning” [tiab] OR “artificial intelligence” [tiab] OR “random forest” [tiab] OR “gradient boosting” [tiab] OR “XGBoost” [tiab] OR “support vector” [tiab] OR “neural network” [tiab] OR “deep learning” [tiab]) AND

(“pharmacometric*” OR “pharmacokinetic*” OR “pharmacodynamic*” [tiab] OR “pharmacogen*” [tiab] OR “drug concentration” [tiab] OR “dose estimation” [tiab] OR “dose optimization” [tiab]) AND

(“2016/01/01” [Date - Publication]: “3000” [Date - Publication]) NOT

(review[Publication Type]).

The search identified a total of 586 articles (as of 30 May 2022), of which 198 were included based on abstract screening. Additional articles were obtained by means of scanning the reference lists of included articles, or by specifically searching in the arXiv database (https://arxiv.org/; accessed from 30 May 2022 until 30 June 2022). Some ML papers are only indexed in pre-print servers, and thus can not be found in PubMed.

## 2. Data Preparation

### 2.1. Data Imputation

Missing data are a frequent occurrence in the clinical setting. When encountering missing data one can drop all data entries or covariates with missing data, impute missing data, or employ maximum likelihood estimation techniques. As many clinical datasets are relatively small, the latter two options are often preferred. Missing data are often categorized in one of three categories; they are either missing completely at random (MCAR), missing at random (MAR; missingness depends on observed data), or missing not at random (MNAR; missingness depends on unobserved data). The source of the missing data can affect the choice of imputation method. In addition, the type of data (i.e., continuous or categorical) can also be a reason for choosing different methods. In the below sections, we will focus on the problem of data imputation, which requires us to choose an appropriate model for the prediction of missing data. How do we select such a model?

#### 2.1.1. Standard Methods for Data Imputation

Imputation can either be performed once (single imputation), or multiple times (multiple imputation). Commonly used methods for single imputation include imputation by mean or mode, grouping missing data in a separate category (in the case of categorical covariates), or regression-based imputation. In multiple imputation, multiple samples are taken from a predictive distribution allowing for the quantification of the resulting variance of model output. This provides a measure of uncertainty of the imputation. A Bayesian multiple imputation strategy has been proposed for NLME models, which presented lower bias of parameter estimates compared to mean value imputation for MCAR and MAR data [10]. A maximum likelihood procedure based on this strategy was also shown to lead to less biased PK parameter estimates compared to mode and logistic regression based approaches [11].

Prior studies have been mainly concerned with the imputation of categorical variables. Model-based (i.e., multiple imputation and likelihood-based) approaches seem to perform well for this kind of data, but do require one to make assumption about the distribution of the data. This can be more difficult for continuous variables. In these cases, it might be compelling to also evaluate regression-based techniques for imputation. Unfortunately, choosing an appropriate regression model when the assumed relationship is non-linear can be difficult. This is especially the case when covariates are correlated. For this reason, ML-based regression techniques have been suggested with the goal of improving the accuracy of regression-based imputation. An early study suggests that when covariates are simulated based on non-linear relationships, the bias of PK parameters after performing imputation can be reduced by using a random forest or neural network prediction model rather than mean imputation [12].

#### 2.1.2. Machine Learning Methods for Data Imputation

A paper by Batista and Mondard compared the accuracy of the *k*-nearest neighbour (*k*-NN) method to mode, decision tree, and rule-based methods for MCAR data imputation [13]. They found that *k*-NN generally was the most accurate method. In *k*-NN, individuals are grouped in *k* clusters based on similarity (for example based on Euclidean distance). Next, missing values can be imputed based on mean/median values from their respective cluster. The method is simple to implement, but might be less effective in small or very homogeneous datasets.

Although Batista and Mondard found that the decision tree-based method was not as accurate, random forest-based approaches have been more successful [14,15,16]. In the popular implementation missForest [14], a random forest is combined with multiple imputation by chained equations (MICE; [17]). MICE is an iterative procedure where each missing covariate is imputed based on the remaining covariates. Initially, missing data are imputed using an arbitrary method (e.g., by their mode) and a model is fit to predict missing data for each covariate independently. This process is repeated with the assumption that each iteration, more accurate imputations of the covariates are used for the predictions. The overall process can be repeated for multiple initial datasets. This way, MICE allows for multiple imputation based on deterministic regression models. Using missForest outperformed *k*-NN and linear MICE for single imputation of MCAR data [14]. However, performing multiple imputation using linear MICE was more accurate than single imputation using missForest on MCAR and MAR data [15]. Performing multiple imputation using missForest led to the overall best accuracy. This implies that multiple imputation procedures might also generally be preferred for regression-based imputation.

Several probabilistic approaches have also been proposed for performing regression-based multiple imputation. Unsupervised deep latent variable models, such as generative adversarial networks (GANs) and variational auto-encoders (VAE), have recently been successfully applied to data imputation problems [18,19]. A GAN is a combination of two neural networks, a generator and a discriminator, which compete against each other. The discriminator learns to discern true from generated data, such that the generator becomes increasingly effective at reproducing real data. The generator learns to represent the generative distribution of the data. The GAIN approach, a GAN specifically developed for the imputation of missing data, was found to more accurately impute MCAR and MAR compared to missForest and MICE [18].

A VAE is a special neural network architecture that learns to encode its input into a distribution of latent variables by means of variational inference. A decoder neural network is then learned to reproduce the original input from samples of this distribution. Mattei and Frellsen describe the combination of an importance-weighted autoencoder with a maximum likelihood objective for imputation of MAR data. This method was found to be more accurate than *k*-NN and missForest [19].

Finally, Gaussian Processes (GPs) are stochastic processes which represent a prior distribution over latent functions. GPs are a non-parametric framework to fit models to data, while simultaneously providing a measure of variance. This allows one to sample from the distribution of latent functions and perform multiple imputation. A recent study has proposed a deep GP approach which suggested more accurate imputation of MCAR data compared to *k*-NN, MICE, and GAIN [20].

#### 2.1.3. Considerations

We have described several ML techniques for data imputation. These techniques might offer improved imputation of covariates that have non-linear relationships with the non-missing covariates. Several findings point to improved performance of regression models when using multiple imputation compared to single imputation. Since most standard regression methods are deterministic, strategies such as MICE are advisable for performing multiple imputation. Using missForest in this context might improve the prediction of more complex covariates. MICE is flexible in that a different model can be used for imputation of each covariate. We can thus use ML methods for the prediction of non-linear covariates while using likelihood-based approaches for categorical covariates. This might be a promising method to explore in the context of NLME modelling.

Recent probabilistic approaches, such as deep latent variable models and GPs, offer an interesting take on regression-based imputation. These methods use likelihood-based approach which might improve imputation accuracy [18,19,20]. However, it is not clear if more accurate methods of data imputation offer significant benefits in terms of reducing the bias of parameter estimates in NLME models. Studies will have to evaluate the benefit of using these more complex methods in the context of pharmacometric modelling.

### 2.2. Dimensionality Reduction

Dimensionality reduction is a technique for detecting patterns in data and reducing this to a lower number of principal components. This can be useful when analysing very high-dimensional data (e.g., gene expression data). Such data are difficult to include in pharmacometric models (and thus rarely are) as their effect might be dependent on a specific combination of patterns. One of the main linear techniques for dimensionality reduction is principal component analysis (PCA). It uses a linear mapping to project each data point to a lower-dimensional representation. The so called principal components are independent and aim to preserve as much of the original variance in the data. Such decompositions can be used to facilitate data visualization and can be used for hypothesis generation (see Section 3). This technique has for example been used to predict the impact of different factor VIII mutations on haemophilia A disease severity [21]. Here, 544 amino acid properties were collected, which they were able to reduce to 19 components using PCA. The researchers could thus drastically reduce data dimensionality while reportedly retaining 99% of the information in the dataset.

Non-linear methods have also been proposed which allow a more flexible mapping to lower-dimensional space. This way, these methods might be able to represent more complex patterns and thus increase the explained variance. One example is the VAE, where the input data are condensed into a set of latent variables. Other prominent examples include uniform manifold approximation and projection (UMAP) [22] and t-distributed stochastic neighbour embedding (t-SNE) [23]. Xiang et al., have recently performed a comparison of ten dimensionality reduction methods for predicting cell type from RNA sequencing data [24]. Although the study shows that no one-size-fits-all method exists, UMAP, t-SNE, and VAE were found to generally outperform other methods of dimensionality reduction. Accuracy of PCA was also high, but it suffered in terms of stability with respect to changes in the number of cells, cell types, and genes.

Another study by Becht et al., compared t-SNE to UMAP to discern cell populations based on patterns in single-cell RNA sequencing data [25]. In their case, UMAP led to more reproducible results and more meaningful visualizations. This is in contrast to the results of Xiang et al., which found t-SNE to be the best performing method [24].

#### Considerations

To our knowledge, the use of dimensionality reduction techniques in the context of pharmacometrics is still quite limited. One of its current principal uses might be as a pre-processing step for generating hypotheses. The lower dimensional representations are ideal for visualization and can be used to detect patterns in otherwise complex data. However, one downside can be that the meaning of the resulting lower dimensional components might be difficult to interpret. In a two dimensional *t*-SNE visualization for example, samples that are closer together and thus more similar would also have been more similar in high dimensional space. The actual features underpinning this similarity can not be discerned from the analysis. In such cases, further inspection of the data would be required to identify the explaining covariates. Another downside is that most of these methods require re-analysis of the data when including new samples. In addition, for each new sample we need to collect the same data to produce the lower dimensional representation.

An important finding is that there might not necessarily be a best method for performing dimensionality reduction [24]. This suggests that different methods should be compared based on the predictive performance of the resulting principal components. However, from our literature search we found that it was more common to use ML to directly learn to predict the outcome of interest from high-dimensional data and select influential covariates for downstream analysis [26,27,28]. We will further discuss such approaches in the context of covariate selection in Section 3. It is unclear how such an approach compares to the above-mentioned methods for dimensionality reduction. More studies are needed to explore the benefit of using these different techniques.

## 3. Hypothesis Generation

### 3.1. Discovery of Patient Sub-Populations

Detecting patient subgroups can help to understand why groups of individuals respond differently to treatment. We found several studies that describe the use of clustering techniques in the context of pharmacometrics. Most focus on grouping patients based on the similarity in terms of treatment response. Kapralos and Dokoumetzidis describe the use of *k*-means clustering for the detection of two patient sub-populations presenting distinctly different absorption patterns of Octreotide LAR [29]. Here, they used the Fréchet distance to define similarity between patients, which can be used to calculate the similarity of longitudinal data in terms of shape. These kinds of measures however do require that all patients are sampled at similar time points. This can be difficult to achieve in practice.

Another study describes the use of mixture models for dividing patients into different classes based on treatment response [30]. Next, the study attempted to predict subgroup based on patient characteristics. This allowed for the identification of clinical indicators that were associated with the different subgroups. The resulting seven treatment classes were validated in an external dataset. This study offers a nice representation how clustering can be used for hypothesis generation and subsequent analysis.

Clustering assumptions can also be implemented within predictive models. In one study, an expectation maximization (EM) approach is used to group individuals based on drug concentration data and a predefined compartment model [31]. Each cluster has its own distinct parameterization and estimate of (residual) variance. This allows us to categorize new patients to a cluster and treat them as were similar patients. Another study has taken an interesting approach where a mixture model-based algorithm is described for grouping individuals into different PK models [32]. These PK models are automatically constructed during the clustering procedure. This approach can thus also be used to generate hypotheses about the appropriate PK models to use for different patient groups. Requirement of a predefined model can also be avoided by combining clustering and supervised ML algorithms. Chapfuwa et al., describe a model for clustering patients in the context of time-to-event analysis [33]. A neural network is used to represent the covariates into a latent variable space, which is made to behave as a mixture of distributions. Each individual is assigned to a cluster in the latent space which contains a corresponding event-time distribution. This allows for the identification of heterogeneous treatment response groups based on covariate data.

#### Considerations

We have discussed examples of studies that have used *k*-means and mixture models to cluster patients in subgroups. Mixture models allow for probabilistic inference, and have been used in more complex model architectures [31,32,33]. These approaches are experimental, but may be of interest to apply to different problems for the purpose of hypothesis generation.

Both mixture models and *k*-means require the user to specify the number of clusters beforehand. This can be difficult when there is no prior information to choose the number of subgroups or when the data cannot be visualized due to high-dimensionality. In those cases, we can either reduce data dimensionality (see Section 2.2), or use some criterion to select the optimal number of clusters [30,34]. An additional downside of mixture models is that they are sensitive to local minima. One study found that prior initialization based on *k*-means++ (an adaptation of *k*-means) allows for a simple procedure to improve model convergence [35].

Clustering patients based on the dependent variable can pose issues in pharmacometric modelling. For example, difficulties arise when clustering patients based on drug concentration measurements when these are collected at different time points. Aside from increasing the speed of processing many samples, the benefit of clustering based solely on the dependent variable might be unclear when differences in drug exposure can be easily discerned from the concentration–time curve. Alternatively, clustering patients based on (individual) PK parameters, summary variables such as area under the concentration time curve (AUC), or independent variables might be more informative in practice.

### 3.2. Covariate Selection

Considering the black-box nature of most ML algorithms, why would one consider using ML for covariate selection? Stepwise covariate modelling (SCM), which is perhaps the most commonly used covariate selection method in pharmacometrics, also has its limitations [36]. Stepwise approaches can lead to difficulties when the data contains many covariates, when there is high collinearity, or when covariate effects are highly non-linear and difficult to determine a priori. The potential of ML-algorithms in this context is that they can be used to learn the optimal implementation of covariates. By performing post-hoc analyses of the model, it can then be possible to determine influential covariates. In the below sections we first discuss limitations of stepwise methods in order to suggest a set of requirements for a successful covariate selection method. Next, we describe ML methods that have been used for this purpose and evaluate if they fit the derived requirements.

#### 3.2.1. Limitations of Stepwise Covariate Selection Methods

In SCM, covariates are included one by one (forward inclusion), and each time the covariate leading to the largest significant decrease in objective function value is included. After all covariates have been tested, the included covariates are removed from the full model one by one (backward elimination). The covariates that do not result in a significant increase in objective function value are removed. This approach leads to some issues. First, due to the potentially large number of statistical tests there is a risk of multiplicity. Second, for an honest implementation of stepwise methods, all hypotheses need to be defined beforehand. This includes all covariates to consider and their functional form. The latter can be quite difficult to determine without first extensively inspecting the data. Finally, the statistical tests are not independent, since the significance of the tests might depend on how and if other covariates have been included. This is especially a problem when there is high collinearity between covariates. Studies have indeed indicated that SCM has a relatively low power when covariates are correlated, have weak effects, or when the number of observations in the dataset is limited [37,38,39].

To reduce the effect of multiplicity and to ensure tests are independent, full model methods are preferred [36]. However, this does not resolve the issue of choosing a suitable functional form to implement each covariate a priori. We suggest the following definition of ideal covariate selection method: it (1) should perform a full model fit (i.e., test multiple hypotheses simultaneously); (2) should be able to learn covariate relationships from data; while (3) penalizing complex solutions (e.g., by regularization); and (4) should allow for the interpretation of resulting relationships. If the method is unable to learn optimal implementations of covariates, we risk making type II errors. If the method does not constrain model complexity it risks inflating the importance of covariates (by fitting arbitrarily complex relationships) resulting in higher type I error. Finally, if the method is not interpretable, we run into problems when actually implementing the selected covariates. If sub-optimal functions are used to implement the selected covariates, they might still result in insignificant effects.

#### 3.2.2. Linear Machine Learning Methods

The least absolute shrinkage and selection operator, or LASSO, is a regression-based method that performs covariate selection by regularization. The LASSO employs the $\ell^{1}$-norm, which penalizes the absolute size of the regression coefficients β. This causes the coefficients of unimportant covariates to be shrunk to exactly zero. All covariates of interest are tested simultaneously in the form of linear equations using a full model fit. Next, a hyperparameter *s*, which controls the size of β such that $\sum_{j=1}^{N_{cov}}|\beta_{j}|\leq s$, can be selected using cross-validation procedures. The use of *s* is a substitution for statistical testing as only the most important covariates will have coefficients greater than zero. The LASSO has seen applications for population PK and Cox hazard models where it outperformed stepwise methods in terms of speed and predictive accuracy [38,40]. Owing to its simplicity, direct integration into the non-linear mixed effects procedure is possible [38].

The LASSO performs a full model fit, penalizes complex solutions, and is interpretable. However, due to the assumption of linear relationships the LASSO fails to meet our second requirement. Since this assumption might not hold for all covariates, there is a risk of type I errors in covariate selection. Although the predictive performance of the LASSO holds up relatively well [41], its performance suffers when the relationship of some of the covariates are non-linear [42].

Multivariate adaptive regression splines (MARS) is a ML algorithm for the approximation of non-linear functions based on piecewise linear basis functions [43]. The method automatically learns the optimal number of splines and their location for single covariates and their combinations. Its classic implementation uses a stepwise approach to prune the number of basis functions to reduce model complexity. Alternatively, a LASSO-based implementation of MARS has been described which presented favourable performance compared to the classic approach [44]. This method has the potential of matching our requirements, but has not yet seen frequent use for the purpose of covariate selection. We have found one abstract mentioning its use, but it did not explore its benefit for approximating non-linear functions [45].

#### 3.2.3. Tree-Based Methods

Tree-based ML algorithms, such as the random forest and gradient boosting trees, have seen recent applications for the purpose of covariate selection. These methods offer a flexible approach to learning non-linear functions, while offering a large number of hyperparameters that can be tuned for regularization. Maximum tree depth, the change in minimum objective function change required for a split, or the minimal number of samples in each node can be empirically set (or automatically using cross validation) to reduce model complexity. The method fits our first three requirements, although the effects of regularization are more difficult to interpret compared to the $\ell^{1}$-norm.

In order to use tree-based methods for covariate selection, covariate importance scores based on “impurity” (also known as Gini importance) or permutation are often calculated. The covariates can be ranked based on these scores. Covariates can be included based on biological plausibility or if they meet a certain threshold [46]. Permutation-based methods are preferred over impurity based methods as the latter can be biased for differently scaled or high cardinal covariates [47]. Simulation studies seem to suggest relative accurate identification of true covariates [42,48].

It is important to note that there is no underlying theory that supports the use of these scores as selection criteria. In addition, another problem is that these scores do not provide information on what functional form to use in order to implement the covariate. It is possible that the relationship underlying the importance has a complicated functional form, and is less important when approximated using basic functions in the final model. As is, this approach does not meet our requirement of interpretability. Novel approaches such as explainable gradient boosting [49,50], might improve the interpretability of tree-based models.

#### 3.2.4. Genetic Algorithms

Genetic algorithms are a special form of search space optimization techniques that rely on evolutionary concepts such as natural selection, cross-over, and random mutation for selecting the most optimal model. They have long been suggested as an alternative approach to model selection for pharmacometric applications [51]. Genetic algorithms allow for testing many opposing hypotheses with respect to model structure simultaneously. In this way, it matches our first requirement. Its direct output is an optimal model (according to the survival function) and matches our fourth requirement.

The general procedure is as follows: first, the full search space is defined, containing all model features to be considered. Next, an objective function is chosen that describes model fitness. Usually this is a combination of the log likelihood of the model and additional penalties for model complexity. Then an initial population is formed containing random combinations of the selected features. For each model, the fitness function is evaluated and the ‘fittest’ models are selected to produce the next generation. This process is repeated for several iterations or when a stopping criteria is met. Since many models have to be fit and evaluated the computational cost of fitting genetic algorithms can be relatively high.

A recent study describes the development of a software-based resource for automating model selection using genetic algorithms, improving their accessibility [52]. This application was compared to stepwise methods and seemed to more accurately recover the true model based on simulated data. Such comparisons are however difficult to make, since the penalty for model complexity was more conservative in the case of the stepwise methods versus the genetic algorithm. The reverse was found in another study, where a stricter fitness function resulted in overly simplified models [53]. Choosing an appropriate fitness function by balancing model accuracy and complexity is not straightforward. It is possible to use heuristic methods such as the Akaike or Bayesian information criterion, but it is likely that there is no one-size-fits-all solution. In addition, the method cannot be used to learn more complex representations of the covariates than were originally included in the search space. Genetic algorithms thus do not meet our second requirement.

### 3.3. Considerations

We have discussed several ML algorithms that can be used for covariate selection. We also proposed four requirements that underlie an ideal covariate selection tool. All discussed methods test all hypotheses simultaneously and match our first requirement. The LASSO offers the most comprehensible approach to regularization but might risk higher type I error due to its assumption of linear relationships. More complex ML algorithms, such as tree-based methods, are more flexible with respect to the representation of non-linear relationships. Perhaps not surprisingly, these methods also suffer the greatest in terms of interpretability (with the exclusion of decision trees). This makes it difficult to translate the results of covariate importance to an appropriate model. The current principal use of tree-based methods might thus be for selecting covariates for subsequent analysis.

The MARS and explainable gradient boosting algorithms come closest to meeting all four requirements. By using piecewise linear functions, MARS approximates non-linear functions and is interpretable. In explainable gradient boosting, a large number of simple models (e.g., small depth decision trees) are fit to each covariate, and relationships can be visualized by summarizing over these models. The visualizations obtained from both methods could be useful in providing an initial intuition about the appropriate functional form to use when implementing covariates. Alternatively, model interpretation methods might be of interest to infer covariate relationships from ML models. We have previously performed an investigation into how one such explanation method can be used to visualize the relationships between covariates and estimated PK parameters [54]. We found that these relationships matched implementations in previous PK models and biological concepts. It might be of interest to further investigate the application of such tools in the context of pharmacometrics. Model explanation methods will be further discussed in Section 5.2.

We have performed a simple simulation study (see Appendix A for implementation details) to showcase the use of some of the previously mentioned methods for covariate selection. Each method was fit to predict individual clearance estimates based on covariate data containing two true covariates and 48 noise covariates. In Figure 1, we depict the measures of covariate importance as determined by means of LASSO, MARS, random forest, or explainable gradient boosting. Each method has correctly identified the two true covariates as important. In addition, we have depicted the approximation of the covariate effect by MARS and explainable gradient boosting (see Figure 1E,F).

We have also discussed the use of genetic algorithms for automation of model selection. Compared to local search or stepwise methods, genetic algorithms offer an intuitive procedure based on evolutionary concepts for simultaneously testing multiple hypotheses with respect to model selection. Software-based resources such as presented by Ismail et al., could help improve accessibility for performing experiments based on genetic algorithms [52]. Although they might be an improvement compared to stepwise methods, genetic algorithms do not meet the suggested requirements for a comprehensive covariate selection method. The main issue lies with selecting an appropriate fitness function. There is no consensus on a generally applicable fitness function. This is worrisome, as choosing an inappropriate fitness function can negatively affect the result.

In summary, none of the presented approaches can meet all our requirements for an ideal covariate selection method. Their purpose might thus mainly be for providing a more informed set of covariates to test. We have mentioned some methods such as MARS and explainable gradient boosting which might also provide intuition about appropriate functional forms to use. Next, genetic algorithms can be used as a full model based approach to testing the hypotheses. More research is however required to optimize this procedure.

## 4. Predictive Models

### 4.1. Machine Learning for Pharmacokinetic Modelling

The development of NLME models is a time-consuming process and requires extensive domain knowledge. Recently, ML algorithms have seen applications as efficient alternatives to NLME modelling [55,56,57,58]. Aside from reducing the time spend on the model building process, ML algorithms can be used as a flexible approach to handle complex and high-dimensional data sources. For example, ML algorithms have been used to directly estimate PK parameters from dynamic contrast enhanced MRI images [59], or to screen almost 2000 genetic markers to find variants affecting tacrolimus exposure [60].

Our search identified many different ML algorithms used for pharmacokinetic modelling. However, we observe that most of these models suffer from problems impeding their reliable use. For example, most models take the current time and drug dose as direct inputs. Aside from leading to issues when multiple drug doses are given, it is uncertain how these inputs will be interpreted by the model. In addition, some models are trained to predict drug concentrations at specific time points, making them unreliable when extrapolating to unseen time points. Finally, since the translation from covariates to drug concentrations can be quite non-linear, these models are prone to overfitting and might require larger datasets in order to generalize well. Based on these issues, we again suggest a set of requirements: (1) the model should be able to produce a continuous solution (i.e., extrapolate to unseen time points); (2) it should be able to adapt to complex treatment schedules (e.g., frequent dosing, mixing different types of administration); (3) it should be able to handle differences in the timing and the number of measurements per patient; and (4) should be reasonably interpretable. Below, we discuss several of the ML algorithms obtained from our literature search and identify two reliable methods for predicting drug concentrations.

#### 4.1.1. Evaluation of Different Approaches

A basic strategy has been to directly predict the concentration-time response based on patient characteristics (e.g., covariates), the dose, and the current time point of interest [55,57]. By predicting a single concentration, we can make independent predictions at each time point per patient. This way, we can meet requirement one and three. In order to satisfy requirement two, we must treat each dosing event as independent and add the remaining concentration from the previous dosing event to the current prediction. A problem with this approach is that the prediction does not represent the total concentration of drug in the body (usually only blood levels) so we lose information about drug accumulation in peripheral tissue. In addition, we assume that the model will learn to predict drug exposure based on the covariates, make adjustments based on the dose, and use the supplied time point to obtain the concentration along the time dimension. It is impossible to completely validate that the model uses these quantities as assumed.

We can however easily show that this approach is unreliable. We performed a simulation study using a neural network to predict real-life warfarin concentrations based on patient age, sex, the dose given at $t_{0}$, and the time point for which to evaluate (see Appendix B for implementation details). The neural network can provide a continuous solution (Figure 2A), and is reasonably able to represent the kinetics of warfarin (e.g., it seems to recognize its absorption and early distribution behaviour). However, when we extend the time frame beyond what was seen during training, we found that the model incorrectly predicts an increase in the exposure (data not shown). In addition, when artificially setting the dose to zero, the model still predicts a response. Admittedly, we can use data augmentation to learn the neural network to predict no exposure when the dose is zero, or when the time point is long after dose administration. We cannot however augment the data with counterfactual cases (specifically with respect to the given dose) and thus the method is inherently unreliable.

Other approaches have used ML to learn optimal dose or AUC instead of a full concentration-time response [61,62,63]. These approaches have their own issues. They will likely be more accurate when measurements are provided as input, resulting in problems relating to requirement three. In addition, it is more difficult to interpret the credibility of the current prediction. In our previous example we could identify problems with the concentration-time curve, but in the case of direct AUC or optimal dose predictions it is more difficult to for example validate the prediction based on visualizations. Interpretation of the model using covariate importance scores can also be difficult [63]. Determination of the AUC or optimal dose based on a prediction of a full concentration-time curve, which can be verified by the observed measurements, will likely still be a more reliable approach.

The next strategy might thus be to use more complex ML algorithms better suited to time-series predictions, such as recurrent neural networks [56,58,64,65]. These studies suggest that these methods can indeed accurately predict the changing drug concentration over time in the context of multiple dosing events [56,58,65]. However, Lu et al., found that such methods did not extrapolate well to unseen dosing schedules [58]. Alternatively, the authors suggest a neural-ODE based approach. Here, a neural network is used to encode the covariates into a latent variable space *z*, which serves as the input to an ordinary differential equation (ODE) solver. This solver is a neural-ODE, a special form of recurrent neural network that learns to represent continuous dynamics similar to an ODE [66]. The neural-ODE is used to explicitly integrate the dosing and timing information. The resulting latent variables adjusted for the current time point are then fed into a decoder network, which produces concentration predictions. They show that this approach does correctly extrapolate to unseen dosing schedules, while also identifying no exposure when the dose is artificially set to zero [58]. This model fits our first three requirements. However, the model architecture is quite complex, and can be difficult to interpret.

We have also recently proposed a similar approach where we directly combine neural networks and ODEs [67]. Here, we also use an encoder network to transform the covariates into latent space variables. In contrast to the neural-ODE approach, we explicitly formulate an ODE system based on compartment models. Dosing events are then used to perturb this ODE system, which directly outputs concentration predictions at the desired time points. This offers several benefits over a neural-ODE: first, by explicitly defining drug kinetics we might reduce the data required to fit the model by imposing explicit constraints. Second, the latent variables now represent PK parameters (e.g., clearance or volume of distribution), which can be compared to previous results. Finally, since we are using a known ODE system, model predictions are credible and interpretable. This way, the method is a better match to requirement four compared to neural-ODE based methods.

#### 4.1.2. Considerations

The last two approaches indicate that ODE-based ML methods are more reliable for the prediction of drug concentrations. In Figure 2, we present a comparison between a naive and ODE-based architecture. We see that only the latter correctly identifies the absence of concentration response when the dose is set to zero (Figure 2B). A neural-ODE can be used to learn the kinetics underlying drug exposure, whereas an explicit ODE system can be used when prior knowledge is available. It is of interest to compare the performance of both these methods.

One remaining opportunity lies with the characterization of prediction uncertainty. In NLME models, the estimation of residual IIV allows for MAP estimation of the PK parameters to correct the prediction based on concentration measurements. Adding this functionality to the above ODE-based models might be of interest for encouraging their adoption in general practice [68]. One approach for obtaining an estimate of predictive and parameter uncertainty are deep ensembles [69]. Here, the predictions of multiple randomly initialized neural networks are combined, and the mean and variance of predictions is presented. This approach is simple to implement and might outperform methods that explicitly estimate parameter uncertainty, such as Bayesian neural networks [70].

Finally, one important aspect of the pharmacometrics pipeline is simulation. As we have shown by removing the dosing event for the neural networks in our example, actively searching for errors in our model is essential for its evaluation. Learning if new patients are different from the data that the model was trained on might help to provide intuition about cases where we trust model output and cases we do not. Simulation is an important approach for facilitating such analyses. We further discuss the topic of model validation in Section 5.2.

### 4.2. Machine Learning for Predicting Treatment Effects

In the wake of -omics research, the interest to personalize patient treatment based on gene or protein expression profiles has increased greatly. High-dimensional data sources stress classical statistical modelling approaches, and many have turned to ML-based approaches [7,26,27]. In addition, some conventional methods such as the cox proportional hazard model, assume linear relationships with covariates. This has prompted the development of tree-based survival models [8], which might be better suited to problems where non-linear interactions can be expected [48]. In the following sections we will focus on the application of ML algorithms for exposure–response modelling (PK/PD models) and survival analysis (time-to-event models).

#### 4.2.1. Exposure-Response Modelling

Exposure-response modelling involves the prediction of treatment effect in relation to the current dose or concentration of an administered drug. It is similar to PK modelling in that it often involves the use of differential equations for describing the dynamics of drug action (e.g., target-site distribution or target binding). Likewise, it is possible to use ODE-based neural network architectures to learn the effects of covariates on model parameters. However, the assumptions underlying the chosen ODE system are often weaker than in the case of PK models [71].

One can discern two types of model components: drug-specific and biological system-specific properties [72]. Drug-specific properties, such as receptor affinity, can be estimated from in vitro data. Biological system-specific properties, such as protein or receptor expression, can only be measured in vivo and can be highly variable between individuals. The latter properties are thus especially sensitive to errors in modelling assumptions. In addition, these effects are often governed by highly non-linear relationships [72]. One approach can be to use neural-ODE based models to learn the relationship between exposure and response from data [73]. Novel interpretable methods have also been described to infer such physical relationships from data [74]. However, in many cases such a direct relationship offers an overly simplistic representation of the biological situation. Alternatively, we can explicitly define part of the ODE system and use neural-ODE to estimate unknown components [75]. This allows the user to explicitly include reasonably certain model components (e.g., drug-specific properties), while neural-ODEs estimates more the more variable and complicated biological system-specific properties from data. It might be of interest to compare such approaches to classical PK/PD approaches.

Novel approaches have also aimed at improving the estimation of biological system-based effects, for example by extrapolating from cell-line data or animal models [76,77,78]. These approaches allow for more frequent measurement of treatment endpoints, and can be used to estimate otherwise difficult to obtain quantities (e.g., spatial distribution of drug in the target tissue [77]). As an example, patient-derived cancer xenograft models can be used to characterize the concentration-dependent effect of drugs on their target based on tumour growth data [76]. Obtaining such results from in vivo patient data would not only be complicated but also undesirable due to the high frequency by with which tumour biopsies would have to be performed.

Adoption of ML algorithms in the context of exposure-response modelling hold exciting opportunities. For example, a recent study describes a method for predicting the drug response of thousands of cancer cell lines based on mutations and expression profiles [79]. Another study describes a method for quantifying the individual variability in tumour dose-response while also identifying important biomarkers [80]. ML techniques can also be used to learn PK or PD (e.g., drug absorption or receptor binding) parameters based on quantitative structure-activity relationships [81]. In short, there have already been many diverse applications of ML in this field, and we expect them to further increase in the future.

#### 4.2.2. Survival Analysis

Time-to-event analysis refers to a set of methods that aim to describe the probability of a specified outcome occurring over time. In the case where only a single event is possible per individual (i.e., survival analysis), non-parametric methods such as the Kaplan-Meijer estimator are used to estimate the distribution describing the proportion of individuals who have ’survived’ over time. These methods allow for the statistical comparison of the efficacy of two competing treatment modalities. Often, we are also interested how covariates affect this efficacy. The standard method for estimating of such effects is the Cox proportional hazard model. Here, the covariates are assumed to affect the hazard in a proportional manner. However, this assumption might be too limiting for a flexible analysis of high-dimensional data, complex time-dependent effects, or multi-state survival models. One might instead turn to ML for learning the effect of covariates or the underlying model structure.

As we have mentioned, the random survival forest model has been proposed for performing non-linear analysis of covariates. The random survival forest was suggested to obtain either similar or lower error compared to Cox proportional hazard models [8]. Recently, deep learning approaches have also been proposed for survival analysis [82,83]. These were generally suggested to obtained higher accuracy (in terms of concordance index) when compared to Cox models and survival forests on several clinical datasets. In addition, these approaches allow for the calculation of the individual risk of prescribing a certain treatment [82]. In Cox models, this risk is constant unless treatment interaction effects are explicitly included, which can be complicated. The neural network-based approach produced treatment recommendations that led to a higher rate of survival compared to random survival forests [82].

Recurrent neural networks have been suggested as a method for estimation of the effect of time-dependent covariates. These methods were again found to outperform previous methods (including neural networks) in terms of concordance index [84,85]. By predicting the individual risk based on current and previous information at discrete time intervals, these methods might improve learning of time-dependent effects.

Multi-state models step away from the usual alive-death dichotomy and instead specify disease progression into intermediary (non-fatal) or competing states [86]. In oncology for example, this allows for the categorization into induction, relapse, remission, and deceased states. This allows for prediction of the risk of relapse following complete remission or survival in the case of relapse [87]. Specification of the dynamics between different states and the influence of covariates might require strong assumptions. A generalizable approach used neural-ODE to learn the likelihood of being in each state over time [88]. This approach obtained improved performance over multi-state Cox models in a competing risk setting.

#### 4.2.3. Considerations

There are many interesting avenues exploring novel applications of ML in exposure-response modelling. The onset of Big Data has resulted in many opportunities for using more advanced and computationally efficient methods for analysing these data. However, some of these tools might still remain at the fringe due to their complexity. Domain-specific reviews providing an overview of recently developed algorithms might help to provide guidelines for optimal strategies for analysis and validation [81]. Without the availability of model code or comprehensive tutorials on the use of complex ML models, adoption of these methods will likely remain limited.

An important consideration for the use of ML for survival analysis is whether the current dataset supports such analyses. Small datasets or those without frequent measurements of the covariates over time might lack the power to correctly describe non-linear effects. As a result, we would recommend evaluating multiple different models for the task at hand. For example, for some datasets, Cox models either performed equal to or better than neural network-based approaches [85]. This could be the case in smaller datasets, or when the data does not support more complex models. In such scenarios, Cox models might be preferred due to their improved interpretability. One might also prefer Cox models when model interpretation is of the highest importance.

## 5. Model Validation

### 5.1. Choosing a Validation Strategy

An essential component of any analysis using ML is a model validation strategy. Arguably, performing model validation is also more generally advisable in the context of pharmacometrics. In contrast to conventional statistical methods however, ML algorithms such as neural networks are extremely flexible. Even neural networks with a single hidden layer are considered to be universal function approximators, meaning that they can fit any data arbitrarily well [89,90]. This flexibility results in a high risk of “overfitting”, a phenomenon where the resulting model is completely tailored to the current dataset such that it generalizes poorly. It is thus important to validate the generalizability of a ML model before it can be used in practice. Arguably the best validation method is to determine the predictive accuracy on independent datasets. Unfortunately, data are often limited. In this section, we report on alternatives for performing model validation.

#### 5.1.1. Options for Estimating Model Generalizability

In the most simple case the dataset is divided in a “train” and “test” set. The train set is used to fit the model, whereas the test set is used to estimate the accuracy of the model. In ML, usually a split using roughly 70–80% of data for training and 20–30% as test data is advised. This is however largely dependent on the size of the test set as it should contain a representative number of samples. Some ML models have additional parameters (i.e., hyperparameters) that can be tuned in order to affect performance. When performing such optimization, the dataset should be split in three parts: a train set (for fitting the model), a “validation” set (for determination of the performance of the current hyperparameters), and a test set (for determination of the accuracy of the final model). A similar approach is be advisable when performing covariate selection.

Performing a single random split of the dataset can be a poor estimate of model generalizability. For this reason, the accuracy is often evaluated on multiple train/test splits and their results are pooled. We will discuss three such techniques for estimating the generalization error: random subsampling without replacement, bootstrapping (subsampling with replacement), and *k*-fold cross validation. A schematic overview of the three methods is provided in Figure 3.

In random subsampling without replacement (also known as Monte Carlo cross validation), the model is fit to a random split of the dataset multiple times, model accuracy is evaluated on the corresponding test sets, and the results are pooled. Since we are sampling without replacement, each sample occurs only once in either the training or test set. Crucially, one should understand that this leads to biased estimates of the true population mean and its standard error. This is because the samples in each split are not independent, and thus violate the Central Limit Theorem. Optionally, one can use the finite population correction factor to improve estimates. However, since the choice of validation strategy is independent of experimental design, possible problems can simply be avoided by performing a bootstrap instead.

In bootstrapping, samples are taken with replacement, resulting in independent samples. Usually *n* (size of the original dataset) samples are taken, and the samples that are not in the training set are used as the test set. Again the results are pooled for a large number of replicates. This estimate can be reliable when all the models converge, which is often not a problem in ML.

Finally, in *k*-fold cross validation, the data are partitioned into *k* “folds”, or subsets, and the model is trained on *k*− 1 folds. The remaining fold is used to estimate test accuracy. Models are fit iteratively so that each fold is used for testing once. *k*-fold cross validation is also frequently performed in the context of hyperparameter optimization.

#### 5.1.2. Considerations

In general, drawing a consensus on the best method for estimating model generalizability is difficult. A downside of random subsampling and bootstrapping is the large computational cost associated with fitting a large number of models. In addition, when sampling with replacement, the number of unique samples in the training set is reduced, which may lead to higher bias of predictions in smaller datasets [91,92]. A similar issue can occur when performing *k*-fold cross validation with a low number of folds [92]. The other extreme, known as leave-one-out cross validation (LOOCV; where k=n), has been suggested to have the best bias-variance trade-off when compared to the other methods [91,92]. Performing ten-fold cross validation leads to similar results compared to leave-one-out cross validation at a lower computational cost [91,92]. The latter is especially relevant as dataset size increases in size (as models have to be fit). Papers have also reported on the inconsistency of LOOCV, specifically that selection of the true data generating model does not actually improve as dataset size increases [93].

Another important consideration when estimating model generalizability is to prevent data leakage. When multiple observations are available per patient, a simple random split might result in different observations of a single individual appearing in both the train and test set. These observations should be grouped to prevent information leakage. Care should also be taken when optimizing hyperparameters. The data that is used to test the current set of hyperparameters should not include samples from the final test set. This means that the dataset should first be divided in a train and test set. The hyperparameters can then be optimized by performing *k*-fold cross validation on data from the train set only. The accuracy of the best model from the cross validation (containing the optimal set of hyperparameters) is then evaluated on the test set. This entire process can also be repeated for multiple test sets, essentially performing an additional (outer) cross-validation. This approach also estimates sensitivity of the hyperparameters to random sub-sets of the data.

Another point to consider is that creating random subsets of the data can exacerbate class imbalances. For example, an algorithm trained to diagnose disease (classifying samples in ‘no disease’ and ‘disease’) can present an inflated accuracy if the model always predicts no disease while the test set does not contain many samples from the disease group. This can often be the case for rare diseases. Alternatively, in case-control studies, train sets can be saturated with control patients, resulting in a model that is unable to make accurate predictions for the case group. In such situations, the data can be stratified so that class proportions are roughly similar in each fold. Care should again be taken to prevent data leakage; data should not be stratified based on the independent variables as this results in more similar train and test sets.

There likely is no single best method of estimating model generalizability. In many cases, *k*-fold cross validation might be preferred when bootstrapping encompasses a too high computational cost. When choosing cross validation, the most important aspect is to choose a suitable value of *k*. Arlot and Celisse provide an excellent survey on model selection using cross validation [94]. Their findings may aid in choosing the appropriate cross validation procedure on a per-problem basis.

### 5.2. Model Interpretation

Explainable artificial intelligence has emerged as an important subfield of ML research. Especially with respect to the adoption of ML for medical applications, understanding why a certain prediction is made is crucial for instilling trust. As an example, model interpretation methods can be used to indicate regions-of-interest underlying predictions for medical image classification [95,96]. There are two types of explanation methods: model-specific and model-agnostic. Model-specific explanation methods for example involve using the regression coefficients from linear models to explain the proportional relationship between covariates and the dependent variable. Although these coefficients have a straightforward meaning, they are not necessarily true; correlated covariates can complicate the estimation of true covariate effects. In more complex models, such as neural networks, the meaning of the model parameters is not immediately obvious. Although neural network specific explanation methods exist [97], generally model-agnostic explanation methods are used. These methods themselves can be considered black-box as they aim to replace the complex model by a more simple, interpretable model.

There have been numerous suggestions for interpretation frameworks aimed at explaining ML model output, including Local Interpretable Model agnostic Explanations (LIME), Deep Learning Important FeaTures (DeepLIFT), and SHapley Additive exPlanations (SHAP) [97,98,99]. An overview of popular methods is provided by Holzinger et al. [100]. This reference also provides short discussions of each method which may provide intuition on what method to use for what goal. It can be difficult to understand how the function of each of these methods affects model interpretation. It is possible that using different frameworks on the same model results in different explanations. It has thus been of interest to find theoretical support for these methods. One method that aims to offer theoretical guarantees is SHAP [99].

SHAP has already seen use in pharmacokinetic modelling. One study used SHAP for the identification of important covariates when using neural networks to predict cyclosporin A clearance [101]. Aside from covariate importance, which only provides a limited interpretation of the model, SHAP can also be used to visualize covariate relationships [54]. To present an example, we performed a SHAP analysis on the prediction of warfarin absorption rate ($k_{a}$) by the previous discussed ODE-based neural network (implementation details in Section B.3). In Figure 4, we depict the relationship between age and $k_{a}$, stratified by sex, as represented by SHAP values. Since we only have a single continuous and categorical variable as input to our neural network we can also obtain their exact functional relationships. In cases when more covariates are included, this is generally not possible. The model predicts a different effect of age on $k_{a}$ for males and females (see Figure 4). The SHAP values allow for the evaluation if the relationships adhere to biological expectations of covariate effects. However, since there are only a few female patients, we should take caution when performing such evaluations. Although the SHAP values seem to be able to represent the effect of the covariates well, extrapolating to unseen samples might be unreliable. Other approaches have been developed in order to estimate the uncertainty of out-of-distribution samples [102]. The use of only a single explanation method might thus not be enough for the complete evaluation of ML models.

#### Considerations

There are many available explanation frameworks for ML models. Here, we have chosen to discuss only one technique to illustrate how these methods can be used. In the case of ODE-based methods, model-agnostic methods are useful to visualize the effect of covariates. However, it is possible that such methods alone are not sufficient for use in clinical practice. It might also be of interest to know if each individual prediction can be trusted, especially when predicting for samples that are different from training data.

Due to the large number of available explanation methods and difficulties with representing their accuracy, choosing the correct method can be a daunting task. In most cases, we can only lean on some theoretical guarantees or expected behaviour on simple examples [103]. Molnar et al., present an excellent overview of pitfalls of these methods and how to resolve them [104]. Since most of the model-agnostic interpretation methods operate in a similar fashion (i.e., they perturb data and evaluate the effect on predictions), they share similar pitfalls. This study also makes the important suggestion that there again is no one-size-fits-all method.

Another great reference is the work by Yang et al. [105]. Here, the authors first outline frequently used concepts in model interpretation, after which they provide two showcases demonstrating how these concepts can be applied for explaining ML model output. This study shows how these techniques can provide insight into the strengths and weaknesses of ML models.

## 6. Main Points

We have discussed several recent applications of ML algorithms in the context of pharmacometrics. More specifically, we have presented how ML techniques can and have been used for data imputation, dimensionality reduction, unsupervised clustering, covariate selection, drug concentration prediction, and treatment response prediction. In general, tree-based models and neural networks were the most frequently used algorithms for these purposes. Most papers report an improvement in performance when comparing the use of these methods to classical approaches. In addition, more complex architectures, which were most frequently based on neural networks, were suggested to be the most accurate. More research is however needed to compare these methods to classical approaches, such as NLME models.

We have started our discussion with the application of ML methods for data preparation. With respect to missing data imputation, the literature suggests lower bias for estimated model parameters when using multiple imputation compared to single imputation. Several ML-based methods have been suggested for imputation of missing data based on regression of observed covariates. It is still unclear however if the added complexity of these methods actually leads to improved estimation of missing data. Evaluation of such methods in the context of the smaller datasets often seen in pharmacometrics is thus required. We have briefly discussed methods for dimensionality reduction. Such approaches might be interesting for facilitating the analysis of complex genetic or proteomics data. However, their benefit compared to using ML to detect influential covariates is not obvious. Although the latter approach can result in the loss of information on covariate dependencies, its results are more easily applicable.

Next, we discussed the application of ML techniques for hypothesis generation. Studies have used unsupervised clustering techniques for the detection of patient subgroups from data. These subgroups can for example be used to generate hypotheses regarding differences in treatment response between patients. ML methods have also been used for the detection of influential covariates. A study has suggested that covariate importance scores produced by the random forest can be used to obtain a better selection of covariates compared to stepwise methods [42]. This and other methods do not yet offer a complete alternative to stepwise methods, but are useful for producing an initial set of hypotheses regarding important covariates to consider for inclusion. Methods for search space optimization such as genetic algorithms are a promising approach for improving the selection of model components that lead to the best performance. This approach requires the selection of a fitness function to control model complexity, which can be difficult to choose. More research is needed for an empirical method for selecting an appropriate fitness function.

ML models have also been used as predictive models in the context of pharmacometrics. We make the point that ODE-based methods outperform other methods in reliability regarding the prediction of drug concentrations. We have showcased this point by using a simple example of how naive methods can misinterpret drug doses when they are passed directly as input. Next, we discuss several ML-based methods for predicting treatment response and efficacy. Again, ODE-based methods show potential for improving prediction reliability, especially in the case of PK/PD modelling. There have also been ML-based approaches for survival analysis. It is the question however if these are appropriate for every analysis, as more complex models might not always result in improved performance.

Finally, we have discussed model validation. Due to the flexibility of many of the discussed methods, deciding on a suitable model validation strategy should be an integral part of the modelling process. The generalization performance of the model is an important metric for judging its appropriateness. Validation of accuracy on a high quality external dataset is often regarded as one of the best options. It is however not clear what is the next best alternative when such data are not available. We would like to urge pharmacologists that are interested in using ML to first consider whether their use case supports the use of these tools. In our experience, we found that imposing constraints on these models (for example based on prior knowledge using ODEs) can help in improving performance when data are sparse. We also want to stress the importance of evaluating what the ML model has learned. Examples include the analysis of the most important covariates, or performing sensitivity analysis (e.g., using methods such as SHAP) with respect to the model parameters. Understanding how the model makes its predictions allows for the removal of any biases, and adapting model regularization to prevent it from making ‘mistakes’. Examination of undersampled regions of the input space can provide insight into the extend to which model predictions can be trusted. Specifically training the model on new data from undersampled patients will help improve generalizability in the long run.

In the coming years, our expectation is that the number of studies exploring the use of ML in pharmacometrics will keep increasing. Perhaps some of the methods mentioned in this review have already become a standard part of the pharmacometrician’s tool kit in the near future. This could be the time for researchers interested in ML to educate themselves in ML concepts and, perhaps, to develop new model architectures better suited to problems in the field of pharmacometrics.

## Figures and Tables

**Figure 1 pharmaceutics-14-01814-f001:**
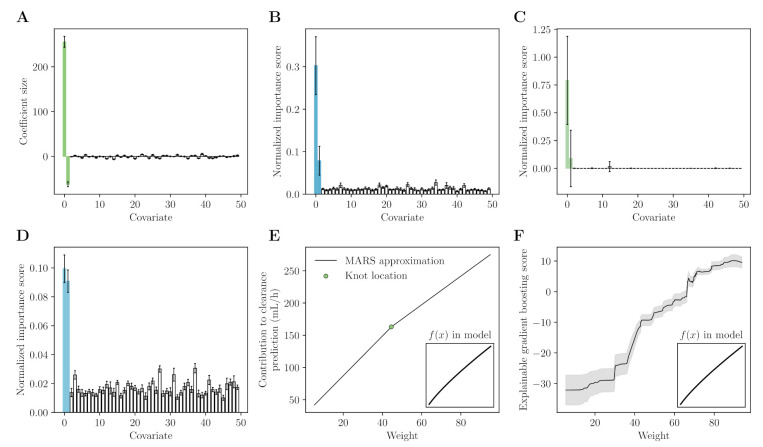
Examples of machine learning-based covariate importance scores. LASSO coefficients (**A**), random forest importance scores (**B**), MARS covariate importance (**C**), explainable gradient boosting scores (**D**), MARS (**E**) and explainable gradient boosting (**F**) approximation of the effect of covariate 1 are shown. Coloured bars indicate true covariates, whereas white bars represent noise covariates. Bar height represents the importance of each covariate. Importance should be larger for true covariates than for noise covariates. The resulting scores can for example be used to select covariates eligible for inclusion in a NLME model. Error bars indicate standard deviation of each score following a ten-fold cross validation. In (**E**), the point indicates the piecewise split location (i.e., a knot). In (**F**), shaded area represents the standard deviation of model predictions in the explainable gradient boosting model. Figure inset represents the function used for covariate 1 in the simulations.

**Figure 2 pharmaceutics-14-01814-f002:**
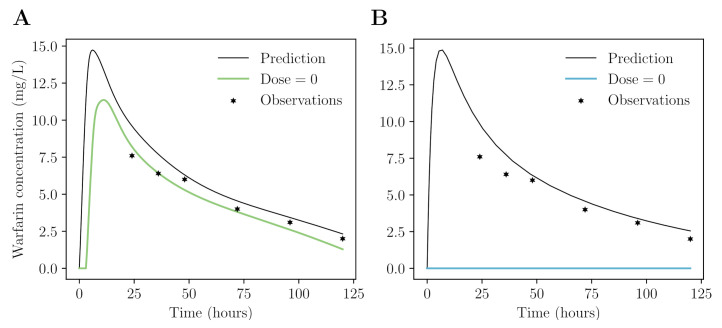
Examples of predicting drug concentrations using neural networks. Concentration-time curves for a single test set patient are shown as predicted using naive (**A**) and ODE-based (**B**) neural networks. Model prediction when artificially setting the dose to zero is depicted by the colored lines. Stars represent the measured warfarin concentrations for the patient.

**Figure 3 pharmaceutics-14-01814-f003:**
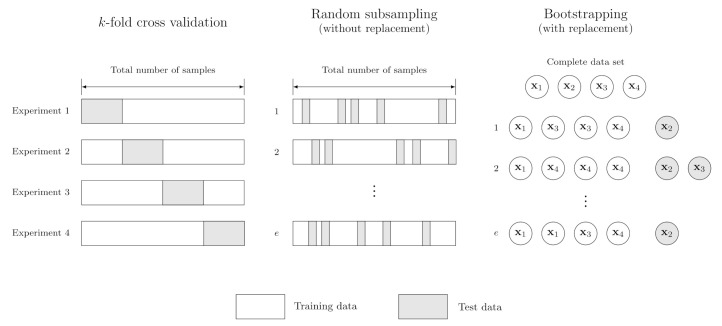
Examples of methods for estimation of model generalization accuracy. Schematic overview of three common validation strategies: *k*-fold cross validation, random subsampling, and bootstrapping (with replacement). The white shapes denote the training data, whereas grey shapes denote testing data. Here, *e* represents the total number of experiments to run.

**Figure 4 pharmaceutics-14-01814-f004:**
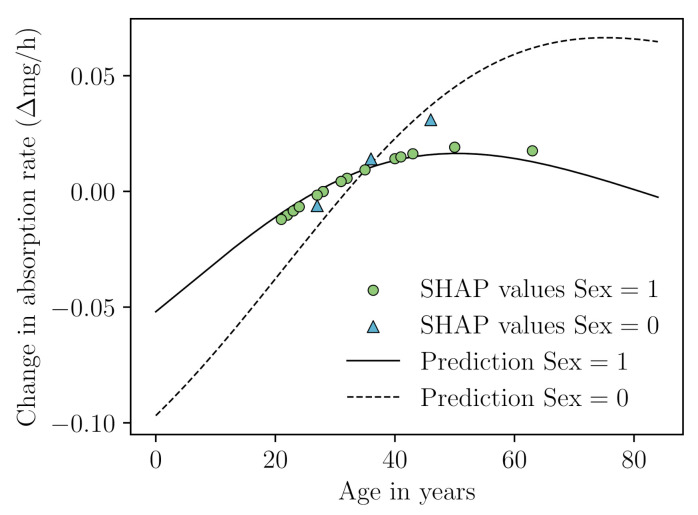
Examples of using SHAP for model interpretation. Change in warfarin absorption rate (Δmg/h) prediction by the neural network as estimated by SHAP values. Here, circles represent the SHAP values calculated for men, whereas triangles represent SHAP values calculated for female patients. Lines represent the neural network predicting when fixing patient sex (solid for male, and dashed for female) and predicting absorption rate based on different values for age.

## Data Availability

All data, including model code, simulated datasets, and the warfarin dataset, are available at https://github.com/Janssena/SI-AIEP-paper.

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
