# Peer review of "Adoption of Machine Learning in Pharmacometrics: An Overview of Recent Implementations and Their Considerations"

_pharmaceutics, 2022, doi:10.3390/pharmaceutics14091814_

Round 1

Reviewer 1 Report

The presented paper is a very nice and comprehensive review about the current situation regarding combination of machine learning and pharmacometrics. It is well written and I cannot even imagine how much time and work it took to collect and review all the presented information. Although there were a few other reviews about this topic published in the last 1-2 years, to my knowledge known of these reviews have such a completeness. Therefore, I recommend to accept this manuscript.

Just a few comments:

The authors seem to intentionally avoid the term artificial intelligence throughout the paper. What is the reason? Usually neural networks are considered to be more related to artificial intelligence than machine learning.

In lines 709-710, one could mention the universal approximation theorem?

Line 857: “where show” sounds strange, maybe demonstrating?

Line 879: “Intend” missing?

Figure 1:  For readers not familiar with the presented methods, panel (A)-(D), are probably difficult to understand. Maybe the authors could try to add a bit more explanation? Additional comments: “Error bars” not “Errors bars”? Brackets for “E” and “F” in the last lines missing?

Author Response

We thank the reviewer for their interest in our work and for taking the time to read our manuscript. We are glad to hear that the reviewer considers our work distinct from previous work. It was our intention to take a more in-depth look at the use of ML, and are glad to hear the reviewer finds the work to be very complete. We have written out responses to the comments of the reviewer. Please note that we have highlighted the actual changes we made in the manuscript both in the updated pdf as well as in our responses:

Comment #1: The authors seem to intentionally avoid the term artificial intelligence throughout the paper. What is the reason? Usually neural networks are considered to be more related to artificial intelligence than machine learning.

Thank you for this question. The reviewer is correct that there is no mention of the term “Artificial Intelligence” in the manuscript (aside from its mention as part of “explainable artificial intelligence”). Machine learning is considered to be a sub-category of artificial intelligence. In our view however, artificial intelligence more specifically refers to algorithms that aim to replicate human intelligence (please note that many other definitions of this term exist, but this is the one we choose to use). Using this definition, we would consider algorithms that are for example used for image classification tasks, natural language processing, or decision making / recommender systems to fall under the umbrella of artificial intelligence. It so happens to be that neural networks dominate in these tasks and so are often considered to represent artificial intelligence.

We felt that the purpose for which the discussed algorithms are used in pharmacometrics is mainly related to statistical learning. We have thus chosen to refer to these tools as machine learning algorithms. Also, since machine learning is a sub-form of AI, we use a more specific definition of the algorithms this way. We do recognize that neural networks are somewhat unique with respect to other algorithms in that they can differ widely in terms of architecture. When specifically referring to neural network based methods we thus use the term “deep learning”, which we consider to have a more specific and different meaning compared to artificial intelligence.

We could write a passage about our interpretation of these definitions, but since the manuscript is already quite long we have not. We ask the reviewer if they agree with the above reasoning for not using the term artificial intelligence.

Comment #2: In lines 709-710, one could mention the universal approximation theorem?

We agree with the reviewer that it is useful to use a more common term to refer to the flexibility of neural networks. We have changed the paragraph as follows:

Original:

“In contrast to conventional statistical methods however, ML algorithms like neural networks are extremely flexible and can fit any data arbitrarily well [88]. This flexibility results in a high risk of "overfitting", ...” line 708-710

Into:

“In contrast to conventional statistical methods however, ML algorithms like neural networks are extremely flexible. Even neural networks with a single hidden layer are considered to be universal function approximators, meaning that they can fit any data arbitrarily well [88, 89]. This flexibilty results in a high risk of "overfitting", ...” line 709-712

In addition we added a reference with proof of the universal approximation capabilities of neural networks:

  • Hornik, K.; Stinchcombe, M.; White, H. Multilayer feedforward networks are universal approximators. Neural networks 1989, 2, 359–366.

Comment #3: Line 857: “where show” sounds strange, maybe demonstrating?

Thank you for this correction. The original text contained an error, and we have replaced

“... provide two showcases where show how these concepts...” line 857

With

“... provide two showcases demonstrating how these concepts...” line 857

As suggested by the reviewer.

Comment #4: Line 879: “Intend” missing?

In the submitted manuscript, line 879 reads as follows:

“We also shortly discussed methods for dimensionality reduction. Such approaches ... [line 880] might be interesting for facilitating the analysis of complex genetic or proteomics data.”

We are unsure what the comment of the reviewer is exactly referring to. Nonetheless we have changed this line as follows:

“We have briefly discussed methods for dimensionality reduction. Such approaches ... ” line 879.

In addition, we changed the following passage to make it more readable:

“Such approaches might be interesting for facilitating the analysis of complex genetic or proteomics data. The benefit of these approaches versus using ML to detect influential covariates is not obvious. Although the latter approach can result in the loss of information on covariate dependencies, its results are more applicable. “ line 879-883

To

“Such approaches might be interesting for facilitating the analysis of complex genetic or proteomics data. However, their benefit compared to using ML to detect influential covariates is not obvious. Although the latter approach can result in the loss of information on covariate dependencies, its results are more easily applicable. “ line 879-883

Comment #5: Figure 1:  For readers not familiar with the presented methods, panel (A)-(D), are probably difficult to understand. Maybe the authors could try to add a bit more explanation? Additional comments: “Error bars” not “Errors bars”? Brackets for “E” and “F” in the last lines missing?

Thank you for this comment. We have indeed expanded on the explanation of the results presented in the figure:

“Figure 1. Examples of machine learning-based covariate importance scores.

LASSO coefficients (A), random forest importance scores (B), MARS covariate importance (C), explainable gradient boosting scores (D), MARS (E) and explainable gradient boosting (F) approximation of the effect of covariate 1 are shown. Coloured bars indicate true covariates, whereas white bars represent noise covariates. Bar height represents the importance of each covariate. Importance should be larger for true covariates than for noise covariates. The resulting scores can for example be used to select covariates eligible for inclusion in a NLME model. Error bars indicate standard deviation of each score following a ten-fold cross validation. In (E), the point indicates the piecewise split location (i.e. a knot). In (F), shaded area represents the standard deviation of model predictions in the explainable gradient boosting model. Figure inset represents the function used for covariate 1 in the simulations.”

Reviewer 2 Report

This review paper has an objective to explain how ML algorithms has been applied into pharmacometrics, especially data preparation, covariate selection and predictive modeling steps. The paper across whole part was written and summarized well, therefore I suppose that many of possible readers and researchers will be able to inspire and get the knowledge from this paper. This reviewer recommend to editorial office for acceptance after minor revision as below:
(Every points that this reviewer pointed out have been marked with highlight option on the attached paper)

[Abstract] 
The last part of affiliation should be deleted. (Current address: affilication 3)

Line: 8~9: The sentence should be better to change from '...pharmacometrics pipeline: data preparation, covariate selection, and predictive modeling ' to 'Data preparation, hypothesis generation, predictive models and model validation' to synchronize the order of main text. 

[Figures]
Every figures should be relocated on suitable position for easy understanding of paper. 

[Author contributions and so on]
Those sections looks like not updating from the template. Author have to clarify those sections before publications as well. 

Author Response

We thank the reviewer for taking their time to read through the entire manuscript, and are glad to hear that they consider it a good summary of the use of ML in pharmacometrics. One of our main goals was to give the common pharmacometrician knowledge and understanding of how ML algorithms have been used, and what approaches might be more succesfull than others. We are happy to hear that the reviewer finds our manuscript to offer knowledge and inspiration in this regard. Below we offer our responses to the comments of the reviewer. Please note that we have highlighted the actual changes we made in the manuscript both in the updated pdf as well as in our responses:

Comment #1: The last part of affiliation should be deleted. (Current address: affilication 3)

Thank you for this comment, we have removed this part from the affiliation section.

Comment #2: Line: 8~9: The sentence should be better to change from '...pharmacometrics pipeline: data preparation, covariate selection, and predictive modeling ' to 'Data preparation, hypothesis generation, predictive models and model validation' to synchronize the order of main text. 

We agree with the reviewer and have changed the corresponding lines in the abstract as follows:

“The goal of this work is to discuss how ML algorithms have been applied in three stages of the pharmacometrics pipeline: data preparation, covariate selection, and   predictive modelling. We will also discuss model validation and considerations    before the use of ML algorithms with respect to each topic.”

To

“The goal of this work is to discuss how ML algorithms have been applied in four stages of the pharmacometrics pipeline: data preparation, hypothesis generation, predictive modelling, and model validation. We will also discuss considerations before the use of ML algorithms with respect to each topic.”

Comment #3: Every figures should be relocated on suitable position for easy understanding of paper. 

Thank you for this comment. Since we are using the LaTeX template from MDPI all figures are collected in a specific section of the manuscript. For the actual publication version of the manuscript, we are operating under the assumption that all figures will be organized in the text by the editor. This is why we have not yet done so.

Comment #4: Those sections looks like not updating from the template. Author have to clarify those sections before publications as well

Thank you for this comment. We have provided this information in the online web form during submission. Since we had already supplied the information we did not think of also providing it in the actual manuscript pdf file. However, we now realize this might mean that the reviewers do not have access to this information at the time of review. We have filled in the information in the sections accordingly.